# Decreased Glucocorticoid Signaling Potentiates Lipid-Induced Inflammation and Contributes to Insulin Resistance in the Skeletal Muscle of Fructose-Fed Male Rats Exposed to Stress

**DOI:** 10.3390/ijms22137206

**Published:** 2021-07-04

**Authors:** Abdulbaset Zidane Shirif, Sanja Kovačević, Jelena Brkljačić, Ana Teofilović, Ivana Elaković, Ana Djordjevic, Gordana Matić

**Affiliations:** Department of Biochemistry, Institute for Biological Research “Siniša Stanković”—National Institute of Re-Public of Serbia, University of Belgrade, 11000 Belgrade, Serbia; shirif.zidane@gmail.com (A.Z.S.); sanja.kovacevic@ibiss.bg.ac.rs (S.K.); brkljacic@ibiss.bg.ac.rs (J.B.); avasiljevic@ibiss.bg.ac.rs (A.T.); ivana.elakovic@gmail.com (I.E.); djordjevica@ibiss.bg.ac.rs (A.D.)

**Keywords:** fructose, chronic stress, skeletal muscle, glucocorticoids, insulin, lipid metabolism

## Abstract

The modern lifestyle brings both excessive fructose consumption and daily exposure to stress which could lead to metabolic disturbances and type 2 diabetes. Muscles are important points of glucose and lipid metabolism, with a crucial role in the maintenance of systemic energy homeostasis. We investigated whether 9-week fructose-enriched diet, with and without exposure to 4-week unpredictable stress, disturbs insulin signaling in the skeletal muscle of male rats and evaluated potential contributory roles of muscle lipid metabolism, glucocorticoid signaling and inflammation. The combination of fructose-enriched diet and stress increased peroxisome proliferator-activated receptors-α and -δ and stimulated lipid uptake, lipolysis and β-oxidation in the muscle of fructose-fed stressed rats. Combination of treatment also decreased systemic insulin sensitivity judged by lower R-QUICKI, and lowered muscle protein content and stimulatory phosphorylations of insulin receptor supstrate-1 and Akt, as well as the level of 11β-hydroxysteroid dehydrogenase type 1 and glucocorticoid receptor. At the same time, increased levels of protein tyrosine phosphatase-1B, nuclear factor-κB, tumor necrosis factor-α, were observed in the muscle of fructose-fed stressed rats. Based on these results, we propose that decreased glucocorticoid signaling in the skeletal muscle can make a setting for lipid-induced inflammation and the development of insulin resistance in fructose-fed stressed rats.

## 1. Introduction

The modern lifestyle is commonly challenged by unpredictable stressful events and the consumption of high-calorie processed foods rich in fructose. Animal and human studies have shown that both chronic stress [1,2] and excessive fructose consumption, particularly in the form of sweetened beverages [3,4], correlate with the development of insulin resistance and related disorders. Therefore, it is of great importance to investigate how these factors, which are present in our everyday lives, influence metabolic health.

Skeletal muscle is the central organ responsible for insulin-regulated glucose uptake and adjustment of whole-body metabolism to energy challenging conditions [5,6]. Resistance to insulin in skeletal muscle is often followed by systemic hyperinsulinemia, dyslipidemia, hypertension and cardiovascular abnormalities comprising the main indicators of metabolic syndrome [7]. Interestingly, studies have shown that insulin resistance in skeletal muscle can occur before the onset of hyperglycemia, even years before a detectable dysfunction of pancreatic β-cells [6]. This is why it is considered that insulin resistance in skeletal muscle is one of the earliest and most important deficiencies in the chain of events that lead to type 2 diabetes (T2D) and metabolic syndrome [6,8].

Aside from carbohydrates, skeletal muscles utilize lipids as an important energy substrate. However, lipid overload and increased influx of free fatty acids (FFAs) into the skeletal muscle can impair insulin signaling [9]. High levels of lipid metabolites, such as fatty acyl-CoAs, diacylglycerols and ceramides have been shown to activate intracellular kinases that promote inhibitory serine/threonine phosphorylation of insulin receptor (IR) and/or insulin receptor substrate 1 (IRS1), impairing insulin action in the muscle [10]. Excessive consumption of fructose, as a highly lipogenic sugar, usually results in the accumulation of triglycerides and FFAs in different tissues [11]. It has been shown that prolonged exposure to a fructose-enriched diet interferes with the activation of IRS1 and Akt resulting in the inhibition of the insulin signaling pathway [4,12]. Furthermore, studies have reported that different types of stressors can induce dyslipidemia in animals and humans [13,14]. Stress can also enhance the increase in plasma triglycerides and prolong their clearance after a meal, causing postprandial lipidemia, a condition closely associated with T2D and metabolic syndrome [14].

In response to lipid and cytokine excess, skeletal muscles have the ability to express proinflammatory molecules and develop inflammatory reactions [8]. Both in vivo and in vitro studies have shown that the proinflammatory cytokine, tumor necrosis factor α (TNFα), can induce inhibitory phosphorylation of IRS1 and decrease the stimulatory phosphorylation of Akt kinase, producing insulin resistance in the skeletal muscle [15,16]. TNFα knockout mice are protected from insulin resistance even in the genetically and diet-induced animal models of obesity [17]. A high-fructose diet has been shown to increase TNFα and the proinflammatory transcription regulator nuclear factor κB (NFκB) in the skeletal muscle of mice and rats [12].

Inflammation is considered an important link between stress exposure and metabolic diseases [18]. Various organs, including the brain, liver, adipose tissue and skeletal muscles, are targets for glucocorticoids, which together with insulin, are crucial in the regulation of glucose and metabolic homeostasis during stress [19]. Taking into account that skeletal muscle represents one of the largest organs in the body responsible for glucose utilization, and therefore is an important regulator of systemic insulin sensitivity, it becomes of interest to understand the role of glucocorticoids in this tissue. Both active and inactive glucocorticoids can enter skeletal muscle from the circulation. Upon entry to muscle cells, the effects of active glucocorticoids (cortisol in humans and corticosterone in rodents) are mediated by binding to the glucocorticoid receptor (GR), a hormone-activated transcriptional regulator. In the cytoplasm, inactivated GR is associated with a chaperone complex consisting of several different molecules including the FK506 binding protein (FKBP51) [20], which is characterized by several laboratories as a strong inhibitor of GR function [21,22,23]. Upon hormone binding, GR dissociates from the chaperone protein complex and undergoes a conformational change which allows its translocation to the nucleus, where it regulates transcription of GR responsive genes [24].

Inactive glucocorticoids (cortisone in humans and 11-dehydrocorticosterone in rodents) are enzymatically activated by 11β-hydroxysteroid dehydrogenase type 1 (11βHSD1) that uses reduced NADPH generated by the hexose-6-phosphate dehydrogenase (H6PDH) in a process called the prereceptor metabolism [25]. The effects of glucocorticoids are mediated primarily by the GR, but the role of mineralocorticoid receptor (MR), which binds glucocorticoids with a higher affinity than GR, should not be neglected. Namely, MR selectivity for aldosterone largely depends on 11βHSD2, an enzyme responsible for the conversion of glucocorticoids to their inactive metabolites unable to bind MR. Although some recent studies confirmed the presence of both MR and 11βHSD2 in the muscles supporting the view that MR functions as a nuclear hormone receptor in this tissue, its function is not well described [26,27]. However, this co-expression of 11βHSD2 and MR in most cells (including myocytes) when combined with the ubiquitous expression of GR, points to the conclusion that GR is the main target of glucocorticoids.

Besides inhibition of muscle glucose utilization, stimulation of protein degradation and reduction of protein synthesis [28], glucocorticoids are well known by their immunomodulatory properties, which are accomplished by GR inhibitory interactions with proinflammatory transcriptional factor NFκB and with AP1 [29], or by transcriptional regulation of DUSP1 phosphatase [30]. The balance between glucocorticoid and proinflammatory signaling is the crucial factor in maintaining adequate immune response and tissue insulin sensitivity.

Lipid species excess and metabolic inflammation caused by caloric overload represent a potent trigger for skeletal muscle insulin signaling deterioration. In addition, exposure to stress could induce lipid metabolism dysregulation and alteration in glucocorticoid activation and signaling that challenge proinflammatory response and maintenance of glucose homeostasis. Although most diabetic patients report higher levels of stressful events than the subjects with normal glucose tolerance, and although chronic stress exposure is usually accompanied by a fructose-enriched diet in everyday life, these two factors are mostly studied separately. Having in mind that skeletal muscle is an important organ for whole-body glucose homeostasis and that metabolic disturbances in the muscle could be one of the earliest defects leading to T2D, the aim of this study was to elucidate whether the combination of excessive fructose consumption and chronic unpredictable stress disturbs insulin signaling in the skeletal muscle of male rats through lipid-induced inflammation and changes in glucocorticoid signaling as a potential underlying mechanism for muscle insulin resistance.

## 2. Results

### 2.1. Lipid Metabolism

In order to evaluate the effects of increased fructose consumption, chronic stress exposure and its combination on lipid metabolism we analyzed the expression of genes involved in FFA transport into the cell, lipolysis and β-oxidation. Two way ANOVA showed the main effect of fructose on FATP1 (F = 10.793, *p* < 0.01) and ATGL (F = 7.581, *p* < 0.01), effect of stress on FATP1 (F = 10.967, *p* < 0.01) and LPL (F = 4.640, *p* < 0.05) and effect of the combination of treatments on FATP1 (F = 11.209, *p* < 0.01) and LPL (F = 9.523, *p* < 0.05). The transcription levels of LPL and FATP1, the genes responsible for triglyceride hydrolysis and FFA transport, were increased in stressed rats maintained on a fructose diet in comparison to all other experimental groups (*p* < 0.05 and *p* < 0.001, respectively) (Figure 1A,C). Additionally, stressed rats maintained on a fructose diet, when compared to untreated animals, exhibited increased expression of the ATGL gene that encodes for the rate-limiting lipolysis enzyme (Figure 1B, *p* < 0.05).

The effect of the fructose diet and stress was examined on the levels of PPARα and δ, lipid-sensing transcriptional regulators with an essential role in the activation of fatty acid uptake and β-oxidation, as well as on the level of CPT1b, an enzyme crucial for FFA transport into the mitochondria and subsequent β-oxidation. Results showed increased protein level of PPARδ (Figure 2A), and increased mRNA levels of PPARα (Figure 2B) and CPT1b (Figure 2C) in stressed animals maintained on a fructose diet (*p* < 0.05, PPARα, PPARδ, CPT1b compared to untreated and *p* < 0.05, PPARα compared to fructose-fed unstressed animals) as a result of the effect of stress (F = 8.743, *p* < 0.01 for PPARα; F = 9.502, *p* < 0.01 for PPARδ; F = 4.518, *p* < 0.05 for CPT1b).

### 2.2. Inflammation

The inflammatory status of the muscle was determined by measuring the relative changes in the protein levels of transcription factor NFκB, its inhibitory phosphorylation on Ser536 [31], NFκB inhibitor IκB, and the expression of proinflammatory cytokines IL-1β, IL-6 and TNFα, which are under transcriptional regulation by NFκB. While the levels of IκB and pNFκB-Ser^536^ proteins remained unaltered (Figure 3A,B), the level of NFκB was increased (*p* < 0.05) and the ratio of pNFκB-Ser^536^ to NFκB was decreased in both stressed groups (*p* < 0.05) in comparison to untreated animals (Figure 3A) as a consequence of stress effect (F = 17.08, *p* < 0.001 and F = 13.24, *p* < 0.01, respectively). Two-way ANOVA of the qPCR results revealed a significant effect of fructose (F = 4.776, *p* < 0.05) and combination of fructose and stress (F = 10.366, *p* < 0.01) on IL-1β as well as the effect of stress (F = 32.29, *p* < 0.0001) on TNFα. Post hoc test showed an increase in the relative levels of TNFα (Figure 3C) and IL-1β (Figure 3D) mRNAs in the muscle of the stressed group maintained on a standard diet as compared to the control (*p* < 0.05), as well as an increase in TNFα mRNA in stressed rats maintained on a fructose diet as compared to untreated (*p* < 0.05) and fructose-fed unstressed animals (*p* < 0.001). Additionally, the decrease in IL-1β mRNA in fructose-fed stressed rats compared to stressed rats on standard diet (*p* < 0.01) and no significant changes were observed in the level of IL-6 mRNA (Figure 3E).

### 2.3. Glucocorticoid Signaling Pathway

Effects of excessive fructose consumption, chronic stress exposure and their combination on glucocorticoid signaling pathway was determined by analyzing the plasma and skeletal muscle levels of glucocorticoid hormone corticosterone, mRNA and protein level of 11βHSD1 as the main enzyme responsible for glucocorticoid hormone activation inside the tissue, as well as protein level of H6PDH, an enzyme that generates the cofactor for this reaction. In addition to the prereceptor metabolism of glucocorticoids, we analyzed mRNA and protein level of GR, its strong inhibitor FKBP51 and GR-responsive gene Kruppel Like Factor 15 (KLF15) in the rat skeletal muscle.

Two way ANOVA showed significant effects of fructose (F = 16.536, *p* < 0.001), stress (F = 32.256, *p* < 0.0001) and combination of treatments (F = 15.748, *p* < 0.001) on corticosterone level in plasma while stress had a significant effect (F = 9.689, *p* < 0.01) on skeletal muscle corticosterone level. As shown in Table 1, the post hoc test revealed an increase in plasma corticosterone level in stressed rats on standard diet (*p* < 0.001) and an increase in skeletal muscle corticosterone level in both stressed groups (*p* < 0.05 and *p* < 0.01, respectively) in comparison to the control group. Although the plasma corticosterone level in fructose-fed stressed rats was higher than in the control group, this rise was not statistically significant and the level of corticosterone in this experimental group was decreased compared to the stressed group (*p* < 0.001).

A significant effect of stress was detected on 11βHSD1 mRNA level (F = 34.933, *p* < 0.0001) causing a decrease in both stressed groups compared to the control and fructose groups (Figure 4B, *p* < 0.01). The protein level of 11βHSD1 and H6PDH was unchanged regardless of the applied treatment (Figure 4C,D).

Significant effect of fructose was detected for both mRNA (F = 16.220, p < 0.001) and protein (F = 9.518, *p* < 0.01) levels of GR and KLF15 mRNA (F = 11.641, *p* < 0.001), stress had an effect on FKBP51 protein level (F = 9.403, *p* < 0.01) while the combination of treatments had an effect on KLF15 mRNA (F = 28.513, *p* < 0.000001). Post hoc test demonstrated decreased protein and mRNA GR levels in both fructose-fed groups compared to the control group (Figure 5A and C, *p* < 0.05 and *p* < 0.01) and decrease in KLF15 mRNA level in all experimental groups compared to untreated animals (*p* < 0.0001 for F and S vs. C; *p* < 0.01 for SF vs. C). On the other hand, the protein level of FKBP51 was markedly increased in fructose-fed stressed rats compared to the control (*p* < 0.05) and fructose groups (*p* < 0.01) (Figure 5B).

### 2.4. Insulin Signaling Pathway

Insulin sensitivity was evaluated by two indexes, QUICKI and R-QUICKI. While there was no statistical difference in the value of the QUICKI index between experimental groups, fructose exhibited an effect on the level of R-QUICKI (F = 5.98, *p* < 0.05) causing a decrease in this index in the fructose-fed stressed rats in comparison to stressed rats on standard diet (Table 1, *p* < 0.05).

Skeletal muscle insulin signaling was evaluated at the protein levels of total IRS1, its activating phosphorylation on Tyr632 and inhibitory phosphorylation on Ser307, as well as at the level of total Akt protein level and its activating phosphorylation on Thr308. Two way ANOVA showed the main effect of fructose on total Akt (F = 5.065, *p* < 0.05), pAkt-Thr308 (F = 10.036, *p* < 0.01), the effect of stress on total IRS1 (F = 10.738, *p* < 0.01), pIRS1-Tyr632 (F = 6.536, *p* < 0.05), ratio of pIRS1-Ser307 to total IRS1 (F = 13.040, *p* < 0.01) and its ratio to pIRS1-Tyr632 (F = 6.951, *p* < 0.05) as well as the effect of combined treatment on Akt (F = 6.360, *p* < 0.05) and pIRS1-Tyr632 (F = 5.449, *p* < 0.05). As shown in Figure 6, in fructose-fed stressed rats post hoc test revealed a decrease in the protein level of total IRS1 (Figure 6C, *p* < 0.01) as compared to the control and fructose groups and diminishment of its activating phosphorylation on Tyr632 (Figure 6D, *p* < 0.05) compared to all other experimental groups. There was no change in IRS1 inhibitory phosphorylation on Ser307, but the ratio of pIRS1-Ser307/total IRS1 (Figure 6G) was increased after exposure to the combined treatment in comparison to the control (*p* < 0.05) and fructose groups (*p* < 0.01). The ratio of activating to inhibitory IRS1 phosphorylation exhibited a decreasing trend (Figure 6H, *p* = 0.08) in response to exposure to the combination of fructose and stress. Additionally, PTP1B, a phosphatase that inhibits the insulin signaling pathway, was increased (Figure 6B, *p* < 0.05 as compared to untreated animals) in the fructose-fed stressed rats as a result of the effects of fructose (F = 5.926, *p* < 0.05) and stress (F = 4.944, *p* < 0.05). Finally, as shown in Figure 7, fructose-fed stressed rats displayed a decrease in total Akt protein (Figure 7B, *p* < 0.05 as compared to all other experimental groups) and in its activating phosphorylation on Thr308 (Figure 7C, *p* < 0.05 as compared to the control and fructose group), while the ratio of pAkt-Thr308/total Akt remained unchanged (Figure 7D).

## 3. Discussion

The results of the present study showed that fructose overconsumption in combination with exposure to chronic unpredictable stress disturbed muscle lipid metabolism, which together with reduced glucocorticoid signaling, provided a setting for the development of inflammation and further insulin signaling impairment detected in the skeletal muscle of male Wistar rats.

Most of the studies so far indicate that disturbed lipid metabolism, especially the one that favors lipogenesis and lipid accumulation in the skeletal muscle, is the main contributor to the development of insulin resistance [10]. However, other studies, including our own, are pointing that another significant contributing factor for the dysfunctional insulin signaling could be the disturbance of lipid metabolism in the opposite direction, i.e., in the direction of increased lipolysis and β-oxidation [32,33,34,35]. In fructose-fed stressed rats in our study, we observed increased expression of lipolytic enzyme ATGL, recognized also as an important factor in the prevention of muscle lipid accumulation [36]. As opposed to the adipose tissue, products of skeletal muscle lipolysis are not released into the bloodstream, but instead are used as local substrates for β-oxidation [37]. This goes in line with our observation of drastically increased expression of CPT1b in the same group, suggesting that mitochondria in the skeletal muscle of fructose-fed stressed rats are under high lipid burden. This was further supported by the increased expression of FA transporters LPL and FATP1, and this result, together with our previous observation of increased circulatory FFAs in the same animals [38], points to the stimulated lipid influx into the muscles and increased lipolysis and β-oxidation after the combination of fructose feeding and stress exposure. Finally, the applied combination of fructose feeding and chronic stress also increased protein levels of both PPARα and PPARδ, the lipid-sensing transcription regulators with an essential role in the activation of FA uptake and β-oxidation in several organs, including the skeletal muscle [9,39], especially under conditions of energy stress [40]. It is noteworthy that both PPARα and PPARδ shift skeletal muscle fuel usage from glucose to FAs [40,41] and that PPARα has the ability to decrease FA esterification to triacylglycerol [39], which altogether inhibits triglyceride accumulation and favors β-oxidation in this tissue.

As previously mentioned, disturbed lipid metabolism in the tissues could be a significant factor underlying the development of insulin resistance. Some authors propose direct effects of PPARα and PPARδ on insulin signaling, as in the study by Creeser et al. where PPARδ agonist decreased insulin-stimulated level of pAkt [42], or in the studies by Finck and colleagues where overexpression of PPARα downregulated IRS1 and glucose transporters in mice skeletal muscle and heart [41,43]. It is noteworthy that in our study, R-QUICKI was decreased in fructose-fed stressed rats indicating disturbed systemic insulin sensitivity. We used R-QUICKI in order to more precisely evaluate systemic insulin sensitivity, since it includes the level of fasting FFA (increased in fructose-fed stressed animals [38]), thus improving the discriminatory power in case of mild insulin resistance when obesity is not present [44], which is the case in our study. In addition, Koves et al. have previously shown that lipid overload of muscle mitochondria is crucial for the development of diet-induced glucose intolerance [45,46]. Accumulation of the lipid intermediates from the incomplete β-oxidation after mitochondria lipid overload could indirectly lead to the development of insulin resistance through enhanced inflammation. It has been previously shown that these lipid intermediates can upregulate NFκB activity, thereby stimulating inflammation [47]. Other authors reported that mitochondrial fatty acid oxidation is essential for lipid-induced inflammation in skeletal muscle hence muscle-specific CPT1b knockout mice do not develop inflammation [48] or insulin resistance regardless of the elevated lipid levels [49]. Indeed, our results clearly showed the presence of inflammation in the skeletal muscle of fructose-fed stressed rats, as judged by the observed increase in the NFκB protein level, decreased ratio of inhibitory serine 536 phosphorylation [31] to total NFκB, and increased transcription of TNFα proinflammatory cytokine. However, it has been pointed out recently that disturbed lipid metabolism (causing systemic and tissue FFA elevation) within different models of insulin resistance and obesity is sometimes not sufficient to induce prolonged inflammation [48]. Thus, it is possible that another significant contributing factor for the prolonged inflammation observed in the skeletal muscle of fructose-fed stressed rats could be suppressed glucocorticoid signaling and lack of glucocorticoid anti-inflammatory effects. Although glucocorticoid signaling was also partially affected after separate treatments, it seems that the combination of treatments that caused disturbed lipid metabolism was sufficient to produce enough metabolic pressure for the rise of inflammation that could cause insulin resistance.

Glucocorticoids are the main hormones responsible for the stress response, but also known to regulate both inflammation and glucose metabolism. Previous studies have shown that glucocorticoids can inhibit insulin signaling and induce insulin resistance in different tissues including the skeletal muscle (reviewed [50]). The results of the present study showed that stress exposure increased systemic corticosterone levels in rats on a standard diet, while additional feeding with fructose solution diminished this stress-induced rise of corticosterone. This lowering effect did not actually lead to a hypocorticosteroid state since the corticosterone level in the plasma of the SF group remained unchanged in comparison to the control values (it was actually higher than in the control group, but not statistically significant). Although several theories of the mechanisms of corticosterone lowering were proposed, including hypoactivity or overstimulation of the HPA axis [51,52], enhanced negative feedback by GCs, and the influence of inflammatory mediators [53], it has also been reported that consumption of caloric, palatable food can decrease responsiveness to stress as a result of reduced central activity [54] and/or activation of reward regions in the brain which initiates so-called reward-eating behavior in order to reduce the stress response [55]. However, the corticosterone level in the skeletal muscle was elevated in stressed animals, probably as a consequence of elevated circulatory glucocorticoids. The main tissue-specific regulation of glucocorticoids goes through the glucocorticoid prereceptor metabolism, with 11βHSD1 and H6PDH as the central enzymes in this process. 11βHSD1 is capable of carrying out both 11-oxo-reductase and dehydrogenase activities, thus interconverting inactive cortisone and active cortisol [56]. The direction of its activity depends on the NADP(H), a cofactor provided by the H6PDH, which is thus responsible for the regulation of the 11βHSD1 set-point of activity. It was previously shown that in H6PDH knockout mice, 11βHSD1 was switched to dehydrogenase activity, resulting in local GC inactivation [57]. However, in our study, protein levels of both H6PDH and 11βHSD1 remained unchanged despite the increased corticosterone levels observed locally in the skeletal muscle of stressed rats. Although the 11βHSD1 protein level was unchanged, which could be attributed to the long protein half-life reported in various cell types and models [58], its mRNA level was markedly decreased in stressed rats, probably as a part of protective mechanism against the increased systemic level of glucocorticoids. Interestingly, decreased expression of 11βHSD1 was persistent in stressed rats on a fructose diet as well, even though fructose feeding ameliorated plasma corticosterone levels in these animals. It is noteworthy that dysregulation of 11βHSD1 in metabolic disturbances is tissue-specific [59] and that its activity, not only in the skeletal muscles and adipose tissue in rats, but also in the liver, could be important. Indeed, our previous study demonstrated that in the same animal model, hepatic 11βHSD1 was not changed by stress treatment, while fructose diet led to stimulated glucocorticoid prereceptor metabolism in the liver, although downstream signaling remained unchanged due to increased glucocorticoid clearance by the reductases [60]. However, in the current study on skeletal muscles, fructose-enriched diet decreased muscle mRNA and protein levels of GR, regardless of the stress. Additional analysis of the expression of KLF15 showed decreased mRNA level of this gene directly regulated by the GR [61] in all experimental groups, which was in parallel with decreased GR protein level. However, protein level of FKBP51 was markedly increased only in the skeletal muscle of stressed rats on fructose diet. It has been previously shown that when in complex with FKBP51, GR shows reduced ligand affinity and nuclear translocation, thereby decreasing GR-dependent transcriptional activity [21]. Taken into account increased FKBP51 protein level together with decreased GR protein level and downregulated GR and KLF15 mRNA levels in the skeletal muscles of our stressed rats on fructose diet, it could be proposed that, at least in this experimental group, GR signaling was weakened. Previous studies, including our own, have reported that fructose has the ability to influence 11βHSD1 and GR expression in different organs, both positively and negatively, and can even have a dominant effect over stress [38,62]. It should be kept in mind that glucocorticoids have a wide range of functions in the body and their action in specific organ results from interaction with different signaling pathways, which is probably the reason for different reports of the effects of fructose consumption on the glucocorticoid signaling pathway. Yet, studies on fructose effects on skeletal muscle glucocorticoid signaling are, to our present knowledge, still scarce. Activation of skeletal muscle glucocorticoid signaling and its anti-inflammatory response have been recognized as key events that protect against inflammatory muscle wasting, emphasizing that glucocorticoid signaling in the muscle have crucial role in limiting harmful effects of excess inflammation [63]. Additionally, it has been shown that 11βHSD1 deficiency may cause earlier onset and later resolving of inflammation in mice with induced rheumatoid arthritis [64]. Since glucocorticoids are well known for their role in the suppression of inflammation, decrease of glucocorticoid signalization in the muscle of fructose-fed stressed rats could influence its ability to properly handle inflammatory pathways thus contributing to the persistence of inflammatory signals. Since, our results clearly showed the presence of inflammation in the skeletal muscle of fructose-fed stressed rats, it could be assumed that diminished muscle glucocorticoid signaling observed in our study could provide a setting for the rise and sustainability of excessive, uncontrolled inflammation, which can further cause muscle metabolic disturbances.

Enhanced inflammation was previously linked with muscle insulin resistance [8] and NFκB activation was suggested to be crucial in the onset of diet- and obesity-induced insulin resistance [65]. Additionally, TNFα assumes a major role in the development of muscle insulin resistance in both obesity and metabolic syndrome, as well as in healthy individuals with insulin resistance in skeletal muscle caused by TNFα treatment [66]. In vivo and in vitro studies revealed that treatment with TNFα lowered the protein levels of activated pIRS1 and pAkt in the skeletal muscle [15,16]. In our study, a combination of fructose-enriched diet and chronic stress decreased protein levels of total IRS1 and Akt in the muscle, as well as their activating phosphorylations, pIRS1-Tyr^632^ and pAkt-Thr^308^. It is noteworthy that there are several lines of evidence supporting the possibility that FKBP51 links stress to metabolic function (as reviewed in [67]). Adipose tissue and skeletal muscles are among the tissues presenting the strongest expression of FKBP51 [68]. Interestingly, FKBP51 is a negative regulator of all three isoforms of the serine/threonine protein kinase Akt [69]. In this context, elevated FKBP51 detected in the skeletal muscles of fructose-fed stressed rats in the present study may be another contributor to the observed decrease of Akt signaling. Although IRS1 inhibitory phosphorylation on Ser^307^ was unchanged regardless of the treatment, the ratio of phosphorylated to total IRS1 was increased in skeletal muscle of fructose-fed stressed rats. In addition, in the same animals, a trend of decrease in the ratio of activating to inhibitory IRS1 phosphorylation was observed (*p* = 0.08). Since both IRS1 and Akt are recognized as critical nodes in the insulin signaling pathway [70], and as their depletion inhibits insulin response [71,72], our results suggest that insulin signaling in skeletal muscle of fructose-fed stressed rats was reduced. It is noteworthy that the impairment of insulin signaling was detected in our study only when both factors were combined, although other authors showed diminished insulin signaling in mice skeletal muscle after chronic psychosocial and noise stress [2,73], as well as in the skeletal muscle, liver and adipose tissue of different rodents after fructose diet alone [4,12,62,74]. However, similarly to the skeletal muscle, the combination of fructose diet and exposure to chronic unpredictable stress has been recently shown as necessary to induce this effect on insulin signaling in the hypothalamus [75] and heart [76].

The detected decrease in pIRS1-Tyr^632^ and pAkt-Thr^308^ is in agreement with the highly upregulated PTP1B observed in the muscle of fructose-fed stressed rats. Changes in the relative level of PTP1B protein, which were proven to reflect its enzymatic activity, have been shown to play a major role in the development of insulin resistance in different human and animal models [77,78,79]. Overexpression of PTP1B in muscle cells from healthy subjects decreased stimulatory Akt phosphorylation [79] and caused not only muscle insulin resistance, but also a systemic impairment of insulin sensitivity [77]. In vivo and in vitro studies on both animals and humans showed that reduction of muscle PTP1B enhances insulin sensitivity, as ascertained by an increase in insulin-stimulated pAkt and hyperphosphorylation of IRS1 [66,79]. Furthermore, TNFα was shown to increase *PTP1B* transcription in skeletal muscle [80] and its activity in cultured myocytes and muscles from adult male mice [66]. Therefore, we suggest that enhanced inflammation in the skeletal muscle of fructose-fed stressed rats could have an important role in the development of impaired insulin signaling, either directly, or through upregulation of the PTP1B.

In conclusion, our results show that the combination of a fructose-enriched diet and chronic stress cause muscle lipid disturbances, inflammation and insulin signaling impairment. Although the direct influence of glucocorticoids in the development of muscle insulin resistance has already been reported by many authors, here we propose the idea that their attenuation can also contribute to the detrimental effects of chronic stress exposure and fructose enriched diet on muscle metabolism. Namely, diminished glucocorticoid signaling could make a setting for unrestricted lipid-induced inflammation, which further contributes to the development of insulin resistance (Figure 8).

Taking into account the importance of muscle metabolism in the maintenance of whole-body insulin sensitivity, our study suggests that a combination of fructose-enriched diet and chronic exposure to stress represents a bigger burden for muscle metabolic function and homeostasis than each of the factors separately. Our study raises the important question of how the modern lifestyle, which inevitably includes daily stressful events affecting glucocorticoid hormones, creates a setting for unfavorable effects on our metabolic health.

## 4. Materials and Methods

### 4.1. Materials

Fructose was purchased from Apipek (Bečej, Serbia) and commercial rodent food from Veterinary Institute Subotica, Serbia. Anti-11βHSD1 (ab109554) and secondary anti-mouse and anti-rabbit IgG H&L horseradish peroxidase (HRP)-linked antibodies (ab97046 and ab6721, respectively) were obtained from Abcam (Cambridge, UK), anti-p NFκB-Ser536 (93H1, #3033) was obtained from Cell Signaling Technology (Danvers, MA, USA), while anti-GR (H-300; sc-8992), anti-FKBP51 (sc-13983), anti-H6PDH (sc-67394), anti-IRS1 (E-12; sc-8038), anti-pIRS1-Tyr632 (sc-17196), anti-pIRS-1-Ser307 (sc-33956), anti-Akt (sc-8312), anti-pAkt-Thr308 (sc-16646-R), anti-PTP1B (N-19, sc-1718-R), anti-PPARδ (PPARβ, F-10, sc-74517), anti-NFκB/p65 (C-20; sc-372) and anti-IκB (sc-371) were all from Santa Cruz Biotechnology (Dallas, TX, USA). Anti-β tubulin antibody was WA3 mouse monoclonal antibody raised against bovine brain β tubulin (Dr. Ursula Euteneuer). Immobilon-FL polyvinylidene difluoride (PVDF) membrane was a product of Millipore (Burlington, MA, USA), while the Amersham ECL Western blotting detection kit was acquired from GE Healthcare Life Sciences (Chicago, IL, USA). A Corticosterone EIA kit was obtained from Immunodiagnostic Systems Ltd. (Boldon, UK). TRIzol^®^ Reagent was obtained from Ambion (Austin, TX, USA), RNase-free DNase I from Ferments (Waltham, MA, USA), while RNase-DNase-free water from Eppendorf (Hamburg, Germany). The following products were purchased from Applied Biosystems (Waltham, MA, USA): the high-capacity cDNA reverse-transcription kit, RNase inhibitor, the TaqMan^®^ universal PCR master mix with AmpErase UNG, and the TaqMan^®^ gene expression assay primer-probe mix for: GR (Rn00561369_m1), 11βHSD1 (Rn00567167_m1), KLF15 (Rn00585508_m1), lipoprotein lipase (LPL) (Rn00561482_m1), adipose triglyceride lipase (ATGL) (PNPL2, Rn01479969_m1), IL-1β (Rn00580432_m1), IL-6 (Rn01410330_m1), TNFα (Rn01525859_g1), and hypoxanthine phosphoribosyltransferase (HPRT) (Rn01527840_m1). Power SYBR^®^ Green PCR master mix was purchased from Applied Biosystems, and specific primer pairs for: carnitine palmitoyltransferase 1b (CPT1b): forward (F) 5′-CCAGGCAAAGAGACAGACTTG-3′, reverse (R) 5′-GCCAAACCTTGAAGAAGCGA-3′; fatty acid transport protein 1 (FATP1): F: 5’-CCCAAGTGGATACAACAGGCA-3’, R: 5’-GGTCTAGAAAGAAGAGCCGGTC-3’; peroxisome proliferator-activated receptors-α (PPARα) F: 5′-CGTTTTGGAAGAATGCCAAG-3′ and R: 5′-GCCAGAGATTTGAGGTCTGC-3′ and HPRT: F 5′-CAGTCCCAGCGTCGTGATTA-3′ and R 5′-AGCAAGTCTTTCAGTCCTGTC-3′ from Invitrogen (Waltham, MA, USA).

### 4.2. Animals and Treatment

Male Wistar rats (2.5 months old), bred in our laboratory, were randomly divided into 4 experimental groups (n = 8–9 per group) during the 9-week treatment: control group (C) fed with commercial standard food and drinking water, fructose group (F) fed with the same food and 20% (*w*/*v*) fructose solution instead of drinking water, stress group (S) was fed the same as control group and exposed to the unpredictable sequence of stressors, 1 or 2 per day for the last 4 weeks, and stress + fructose group (SF), which was fed like the fructose group and exposed to stress as the stress group. Fructose concentration, as well as the type, sequence and duration of applied stressors, were chosen to resemble modern human lifestyle [81,82]. Allocation of the animals to the experimental groups was performed by appropriate randomization method in order to ensure blinding and reduction of systematic differences in the characteristics of animals allocated to different groups. The stress protocol was a modified protocol of Joels et al. [81], and included the following daily stressors: forced swimming in cold water for 10 min, physical restraint for 60 min, exposure to a cold room (4 °C) for 50 min, wet bedding for 4 h, switching cages for 2 h, rocking cages for 1 h, and cage tilt (45°) overnight. The number (1 or 2) and type of daily stressor(s), as well as the onset of stress exposure (between 4 and 7 p.m. for the overnight cage tilt, and between 9 a.m. and 4 p.m. for all the other stressors) were randomly selected at the beginning of the treatment. A particular stressor was never applied on two consecutive days or twice a day. All experimental groups had ad libitum access to food and drinking fluid during the treatment period. Detailed composition of the food published previously [60]. Animals were housed three per cage and kept under standard conditions at 22 °C with a 12-h light/dark cycle. Animals had constant veterinary care during the course of the experiment. All animal procedures were in compliance with Directive 2010/63/EU on the protection of animals used for scientific purposes and approved by the Ethical Committee for the Use of Laboratory Animals of the Institute for Biological Research Siniša Stanković, University of Belgrade (permit No. 02-11/14 obtained on 13 November 2014).

### 4.3. Determination of Plasma Corticosterone Level and Systemic Insulin Sensitivity

After overnight fasting, animals were killed by rapid decapitation with a guillotine (Harvard Apparatus, Holliston, MA, USA). In order to prepare blood plasma, trunk blood was collected in EDTA-coated tubes and centrifuged at low speed (1600× *g*/10 min). Obtained plasma was stored at −20 °C until use.

Corticosterone concentrations in the plasma and skeletal muscle were determined by Corticosterone EIA kit according to the manufacturer’s instructions (Immunodiagnostic Systems LTD, The Boldons, UK). Spectrophotometer (Multiskan Spectrum, Thermo Fisher Scientific, Waltham, MA, USA) was used to measure absorbance (at 450 nm and 650 nm) while 4PL curve fitting method (Prism 5.0, GraphPad Software, Inc., La Jolla, CA, USA) was used to calculate corticosterone concentrations.

In order to determine insulin sensitivity, two indexes were used. QUICKI (Quantitative insulin sensitivity check index) and R-QUICKI (Revised quantitative insulin sensitivity check index) were calculated using formulas 1/[log insulin (µU/mL) + log glucose (mg/dL)] and 1/[log insulin (µU/mL) + log glucose (mg/dL) + log NEFA (mmol/L)], respectively. Blood glucose levels were determined by MultiCare strips (Biochemical Systems International, Italia) while plasma insulin concentrations were determined by RIA method, using rat insulin standards (INEP, Belgrade, Serbia). Assay sensitivity was 0.6 mIU/L, and an intra-assay coefficient of variation was 5.24%. The levels of NEFA in plasma were measured colorimertrically (Semi-auto Chemistry Analyzer, Rayito 1904C) using a Randox NEFA kit (Randox Laboratories Ltd., Crumlin, UK).

### 4.4. Preparation of Muscle Tissue Extract

After rapid decapitation with a guillotine, the gastrocnemius muscle was isolated, washed with saline, dried and stored in liquid nitrogen until use. After thawing, muscles were homogenized in ice-cold RIPA buffer 1:4 (*w*/*v*) (50 mM Tris-HCl, pH 7.4, containing 150 mM NaCl, 10 mM EDTA-Na2, 10 mM EGTA-Na2, 0.5% Triton X, 1% NP40, 0.1% SDS, 2 mM dithiothreitol, and protease and phosphatase inhibitors) using Janke-Kunkel Ultra Turrax T25 (IKA^®^, Staufen, Germany). Homogenates were sonificated 3 × 5 s, 1A, 50/60 Hz on ice, then incubated on ice for 30 min with frequent vortexing, and finally centrifuged 20 min on 14,000× *g*, 4 °C. The obtained supernatants were used as the muscles tissue extracts.

### 4.5. SDS Polyacrylamide Gel Electrophoresis and Western Blot

Samples were mixed 1:1 with 2 × Laemmli buffer and boiled for 5 min. Proteins (50 μg) were separated by electrophoresis through SDS polyacrylamide gels and transferred onto the PVDF membrane. To detect proteins involved in lipid metabolism, insulin and glucocorticoid signaling and inflammation, membranes were incubated with appropriate primary antibodies, followed by HRP-conjugated secondary antibodies (1: 30,000). To correct the protein load, membranes were probed with anti-β tubulin primary antibody followed by the respective HRP conjugated secondary antibody. Immunopositive bands were visualized by the ECL reaction and quantified by ImageJ software.

### 4.6. RNA Extraction and Reverse Transcription

Total RNA was extracted from muscles using TRIzol^®^Reagent following the manufacturer’s protocol and dissolved in 30 µL of RNase-DNase free water with RNase inhibitor. The integrity of RNA was confirmed by 1% agarose gel electrophoresis while its concentration and purity were tested spectrophotometrically (OD 260/280 > 1.8 was considered satisfactory). DNAse I treatment was performed according to the manufacturer’s instructions, in order to remove DNA contamination. cDNA was synthesized from 2 µg of RNA. The reverse transcription was performed using High Capacity cDNA Reverse Transcription kit in a 20 μL reaction with MultiScribeTM Reverse Transcriptase in the presence of Random Primers. Reactions were carried out under RNase-free conditions at 25 °C for 10 min followed by 37 °C for 2 h and final denaturation at 85 °C for 5 min. The cDNA was stored at −80 °C until further use.

### 4.7. Real-Time PCR

The expression of GR, KLF15, 11βHSD1, LPL, ATGL and proinflammatory cytokines (IL-1β, IL-6 and TNFα) was analyzed by TaqMan quantitative PCR (qPCR), and the expression of CPT1b, FATP1 and PPARα was analyzed by SYBR^®^ Green qPCR using the AB Prism 7000 and QuantStudio3 sequence detection system. All reactions were performed in 25 μL volume in triplicate, and the mean Ct value for each triplicate was used for further analysis. The TaqMan reaction mix consisted of 1 × TaqMan^®^ universal PCR master mix, with AmpErase UNG, 1 × TaqMan^®^ gene expression assay, and cDNA template (20 ng of RNA converted to cDNA). SYBR^®^ Green reaction mix consisted of 1 × power SYBR^®^ Green PCR master mix, specific primer sets, and cDNA template. Thermal cycling conditions were: 2-min incubation at 50 °C for UNG activation, 10 min at 95°C followed by 40 cycles at 95 °C for 15 s and 60 °C for 60 s. The specificity of the SYBR^®^ Green reaction was verified by melting curve analyses. To detect possible reagent contamination no template control was included for each target gene. Relative quantification of gene expression was performed using the comparative 2–ΔΔCt method. HPRT was used as reference gene.

### 4.8. Statistical Analysis

In order to determine the effects of fructose, stress and their interaction, two-way ANOVA followed by the post hoc Tukey test was used by the statistical program STATISTICA8, for each experiment. A probability level less than 0.05 was considered to be statistically significant. Data are presented as mean ± SEM.

## Figures and Tables

**Figure 1 ijms-22-07206-f001:**
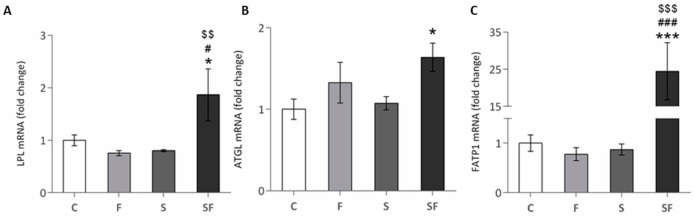
Level of LPL (**A**), ATGL (**B**) and FATP1 (**C**) in skeletal muscle of control (C), fructose (F), stress (S), and stress + fructose (SF). LPL, ATGL and FATP1 mRNA levels compared to HPRT mRNA were determined by TaqMan (LPL and ATGL) and SYBR^®^ Green (FATP1) real-time PCR in the skeletal muscle. The values represent the mean ± SEM (*n* = 8–9 animals per group). All measurements were done in triplicate. Significant between-group differences from post hoc Tukey test are given as follows: * *p* < 0.05 and *** *p* < 0.001, SF vs. C; # *p* < 0.05 and ### *p* < 0.001, SF vs. F; $$ *p* < 0.01 and $$$ *p* < 0.001, SF vs. S. LPL—lipoprotein lipase; ATGL—adipose triglyceride lipase; FATP1—fatty acid binding protein 1; HPRT—hypoxanthine phosphoribosyltransferase.

**Figure 2 ijms-22-07206-f002:**
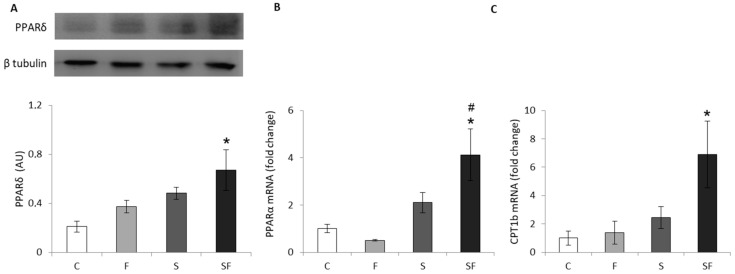
Level of PPARδ (**A**), PPARα (**B**) and CPT1b (**C**) in skeletal muscle of control (C), fructose (F), stress (S), and stress + fructose (SF). Protein level of PPARδ (normalized to β-tubulin) was measured by Western blot in the muscle tissue extracts. The values represent the mean ± SEM. PPARα and CPT1b mRNA levels compared to HPRT mRNA were determined by SYBR^®^ Green real-time PCR in the skeletal muscle. The values represent the mean ± SEM (*n* = 8–9 animals per group). All measurements were done in triplicate. Significant between-group differences from post hoc Tukey test are given as follows: * *p* < 0.05, SF vs. C; # *p* < 0.05, SF vs. F. PPARδ—peroxisome proliferator-activated receptor-δ; PPARα—peroxisome proliferator-activated receptor-α; CPT1b—carnitine palmitoyltransferase 1b; HPRT—hypoxanthine phosphoribosyltransferase.

**Figure 3 ijms-22-07206-f003:**
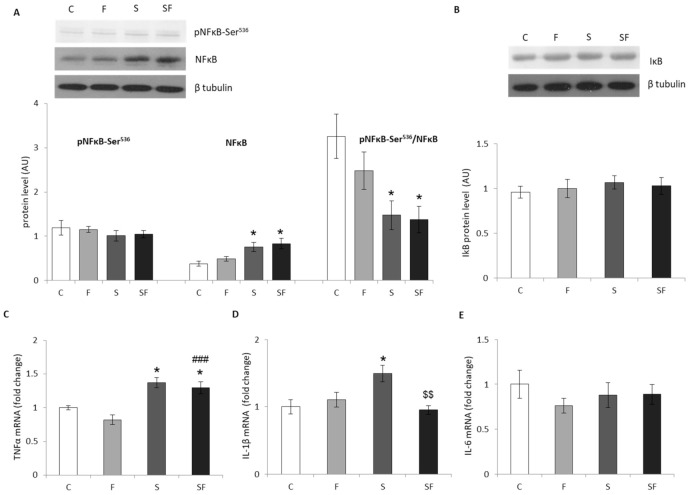
Skeletal muscle inflammatory status in control (C), fructose (F), stress (S), and stress + fructose (SF). Protein levels of NFκB and pNFκB-Ser^536^ (**A**) and IκB (**B**) (normalized to β tubulin) were measured by Western blot in the muscle tissue extracts. The values represent the mean ± SEM. TNFα (**C**), IL-1β (**D**), IL-6 (**E**) and mRNA levels compared to HPRT mRNA were determined by TaqMan real-time PCR in the skeletal muscle. The values represent the mean ± SEM (*n* = 8–9 animals per group). All measurements were done in triplicate. Significant between-group differences from post hoc Tukey test are given as follows: * *p* < 0.05, S, SF vs. C; ### *p* < 0.001, SF vs. F and $$ *p* < 0.01, SF vs. S. NFκB—nuclear factor κB; IκB- inhibitor of NFκB; TNFα—tumor necrosis factor α; IL-1β—interleukin 1β; IL-6—interleukin 6; HPRT—hypoxanthine phosphoribosyltransferase.

**Figure 4 ijms-22-07206-f004:**
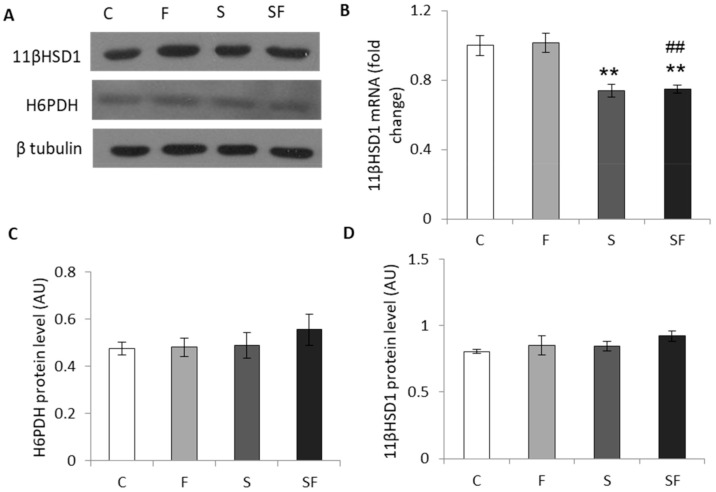
Glucocorticoid prereceptor metabolism in skeletal muscle of control (C), fructose (F), stress (S), and stress + fructose (SF). Representative Western blots (**A**) of protein levels of 11βHSD1 (**D**) and H6PDH (**C**) in the muscle tissue extracts. The values represent the mean ± SEM for each protein normalized to β tubulin expressed in arbitrary units (AU). 11βHSD1 mRNA level (**B**) compared to HPRT mRNA was determined by TaqMan real-time PCR in the skeletal muscle. The values represent the mean ± SEM (*n* = 8–9 animals per group). All measurements were done in triplicate. Significant between-group differences from post hoc Tukey test are given as follows: ** *p* < 0.01, S, SF vs. C; ## *p* < 0.01, SF vs. F. 11βHSD1—11β hydroxysteroid dehydrogenase type 1; H6PDH—hexose-6-phosphate dehydrogenase; HPRT—hypoxanthine phosphoribosyltransferase.

**Figure 5 ijms-22-07206-f005:**
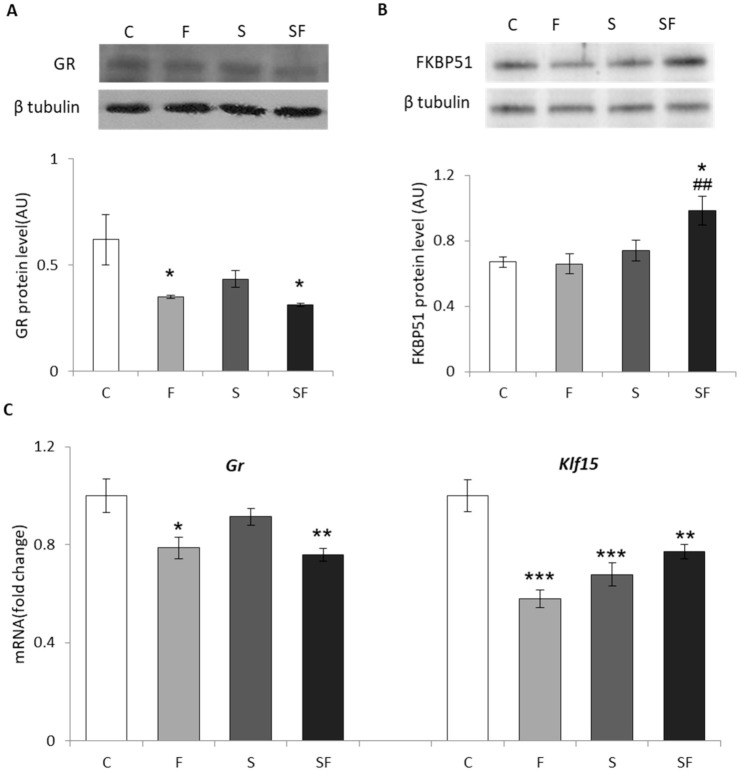
Glucocorticoid receptor signaling pathway in skeletal muscle of control (C), fructose (F), stress (S), and stress + fructose (SF). Protein levels of GR (**A**) and FKBP51 (**B**) were measured by Western blot in the muscle tissue extracts. The values represent the mean ± SEM for each protein normalized to β tubulin expressed in arbitrary units (AU). GR and KLF15 mRNA levels (**C**) compared to HPRT mRNA were determined by TaqMan real-time PCR in the skeletal muscle. The values represent the mean ± SEM (n=8-9 animals per group). All measurements were done in triplicate. Significant between-group differences from post hoc Tukey test are given as follows: * *p* < 0.05, ** *p* < 0.01, *** *p* < 0.001, treated groups vs. C; ## *p* < 0.01, SF vs. F. GR—glucocorticoid receptor; FKBP51—FK506-binding protein 51; KLF15—Kruppel Like Factor 15; HPRT—hypoxanthine phosphoribosyltransferase.

**Figure 6 ijms-22-07206-f006:**
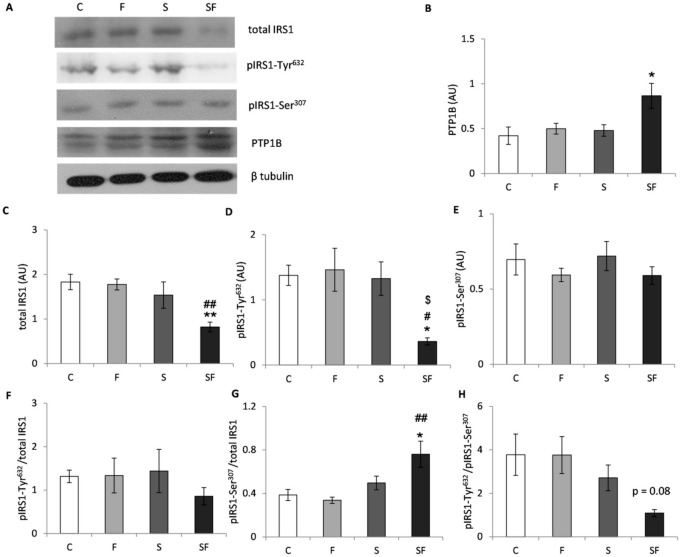
Protein levels of total IRS1, its activating and inhibitory phosphorylations and PTP1B in skeletal muscle of control (C), fructose (F), stress (S), and stress + fructose (SF). Representative Western blots (**A**) of protein levels of total IRS1 (**C**), pIRS1-Tyr632 (**D**), pIRS1-Ser307 (**E**) and PTP1B (**B**) in the muscle tissue extracts. The values represent the mean ± SEM (n=8-9 animals per group) for each protein normalized to β tubulin expressed in arbitrary units (AU) as well as for the ratio of phosphorylated and total protein (**F**,**G**) and ratio of activating and inhibiting phosphorylation (**H**). Significant between-group differences from post hoc Tukey test are given as follows: * *p* < 0.05 and ** *p* < 0.01, SF vs. C; # *p* < 0.05 and ## *p* < 0.01, SF vs. F; $ *p* < 0.05, SF vs. S. IRS1—insulin receptor substrate 1; PTP1B—protein tyrosine phosphatase 1B.

**Figure 7 ijms-22-07206-f007:**
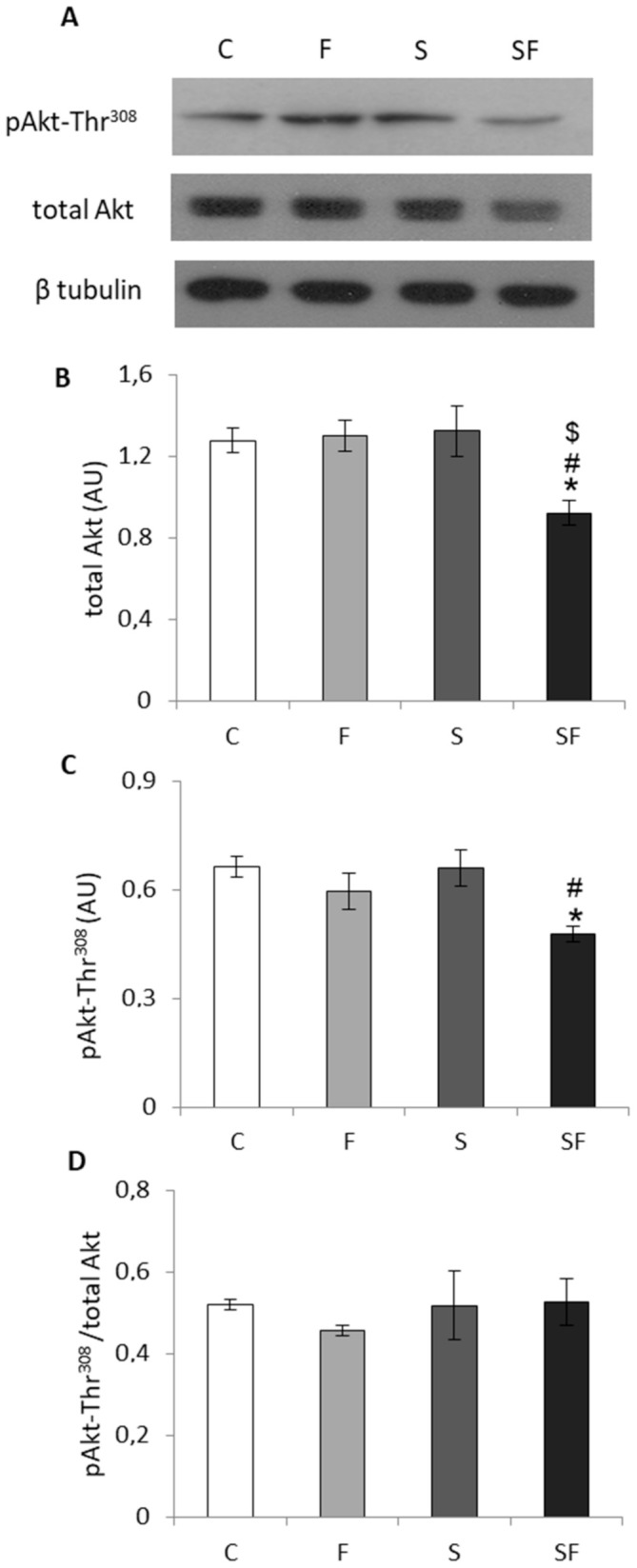
Protein levels of total Akt and its activating phosphorylation in skeletal muscle of control (C), fructose (F), stress (S), and stress + fructose (SF). Representative Western blots (**A**) of protein levels of total Akt (**B**) and pAkt-Thr308 (**C**) in the muscle tissue extracts. The values represent the mean ± SEM (*n* = 8–9 animals per group) for each protein normalized to β tubulin expressed in arbitrary units (AU) as well as for the ratio of phosphorylated and total protein (**D**). Significant between-group differences from post hoc Tukey test are given as follows: * *p* < 0.05, SF vs. C; # *p* < 0.05, SF vs. F; $ *p* < 0.05, SF vs. S. Akt—protein kinase B.

**Figure 8 ijms-22-07206-f008:**
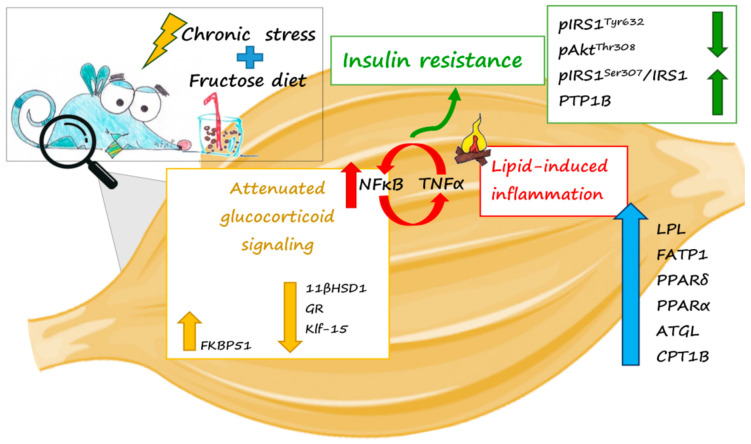
Attenuated glucocorticoid signaling in the skeletal muscle can make a setting for lipid-induced inflammation and the development of insulin resistance in fructose-fed stressed rats. FKBP51—FK506 binding protein, 11βHSD1—11β hydroxysteroid dehydrogenase type 1, GR—glucocorticoid receptor, KLF15—Kruppel Like Factor 15, NFκB—nuclear factor κB; IκB—inhibitor of NFκB; TNFα—tumor necrosis factor α, IRS1—insulin receptor substrate 1, AKT—protein kinase B, LPL—lipoprotein lipase, ATGL—adipose triglyceride lipase, FATP1—fatty acid binding protein 1, PPARδ—peroxisome proliferator-activated receptor-δ, PPARα—peroxisome proliferator-activated receptor-α; CPT1b—carnitine palmitoyltransferase 1b.

**Table 1 ijms-22-07206-t001:** Physiological measurements: Plasma and skeletal muscle corticosterone level and systemic insulin sensitivity indexes in control rats, fructose-fed rats, rats exposed to chronic unpredictable stress, and fructose-fed stressed rats.

	Control	Fructose	Stress	Stress + Fructose	Two-Way ANOVA		
					Fructose	Stress	Interaction
Corticosterone (ng/mL)	114.00 ± 31.81	107.70 ± 16.82	742.20 ± 120.90 ***	219.10 ± 32.29 ^$$$^	*p <* 0.001	*p* < 0.0001	*p* < 0.001
Corticosterone (ng/mg)	0.113 ± 0.024	0.291 ± 0.065	0.356 ± 0.075 *	0.424 ± 0.064 **	NS	*p* < 0.01	NS
R-QUICKI	0.172 ± 0.003	0.169 ± 0.003	0.177 ± 0.004	0.164 ± 0.003 ^$^	*p* < 0.05	NS	NS
QUICKI	0.168 ± 0.003	0.167 ± 0.003	0.174 ± 0.003	0.165 ± 0.002	NS	NS	NS

The data are presented as means ± SEM (*n* = 8–9 animals per group). A value of *p* < 0.05 was considered statistically significant. Significant between-groups differences obtained from two-way ANOVA followed by post hoc Tukey test are given as follows: * *p* < 0.05, ** *p* < 0.01, *** *p* < 0.001, treated animals vs. C; ^$^ *p* < 0.05, ^$$$^ *p* < 0.001, SF vs. S; NS: not significant.

## Data Availability

The data presented in this study are available on request from the corresponding author.

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
