# Peer review of "Decreased Glucocorticoid Signaling Potentiates Lipid-Induced Inflammation and Contributes to Insulin Resistance in the Skeletal Muscle of Fructose-Fed Male Rats Exposed to Stress"

_ijms, 2021, doi:10.3390/ijms22137206_

Round 1

Reviewer 1 Report

See attached file.

Reviewer 2 Report

Dear authors, thank you for the interesting study. There are some points to be discussed.

  1. 11β-HSD1 is a low-affinity NADP(H)-dependent bi-directional enzyme, capable of carrying out both 11-oxo-reductase and dehydrogenase reactions, interconverting inactive cortisone, and active cortisol. Hexose-6-phosphate dehydrogenase (H6 PDH) is the first enzyme in the microsomal version of the cytosolic-based pentose phosphate pathway and appears to be responsible for NADPH provision for 11β-HSD1 thus regulating its set-point of activity. With that, H6 PDH is related to glucose-6-phosphate dehydrogenase (G6 PDH). Moreover, the switch in 11β-HSD1 activity, with the enzyme acting as a dehydrogenase, resulting in local GC inactivation was observed in Hexose-6-Phosphate Dehydrogenase knockout mice (PMID: 21106871).

Thus, there is a question, why NADH/NADPH was not studied and discussed.

  1. Local regulation is considered to be associated with the action of renal 11b-HSD2 without altering circulating levels of cortisol, while 11β-HSD1 acting as a reductase appears to amplify glucocorticoid levels inside expressing cells in adipose tissue and brain, increasing local glucocorticoid levels and is critical to developing the phenotype of GC excess (PMID: 23899562). Based on these, please explain and/or give data of
    1. Why did the concentrations of renal 11b-HSD2/hepatic 11β-HSD1 were not measured
    2. Please tell why only circulating levels of GS were studied. The information of their tissue concentrations could be of interest (PMID: 28445389, PMID: 28765040).
  2. Dysregulation of 11β-HSD1 in obesity is tissue-specific (PMID: 28765040) and thus the determination of its activity not only in skeletal muscles and adipose tissue in rat, but also in liver (where it is decreased) is necessary.
  3. Please tell why the inflammatory markers in the blood (interleukins) were not measured.
  4. Please discuss, why among all the ILs, only TNF and NkF were changed.
  5. Please make the Figure describing the pathophysiology of the processes (scheme) described.
  6. Several distinct theories of the origin of hypocorticsteroid states were proposed during the last decades. Among them are hypoactivity (PMID: 8234641), or overstimulation of the HPA axis (PMID: 15950390), downregulation of CFR receptors at the level of the pituitary gland (PMID: 2178035), enhanced negative feedback by GS, secondary to an increased sensitivity of glucocorticoid receptors in target tissues (PMID: 11495096), and the influence of inflammatory mediators and processes (PMID: 32401103,). Thus, please discuss the mechanism of lowering the circulating levels of GS in more detail. Probably, the inflammatory conditions were the key element of this phenomenon.
  7. Lines 26-27 (Based on these results, we propose that lack of glucocorticoid anti-inflammatory effects). Probably it is better to rephrase this, as, according to the results, the GC levels were not lower than normal (control) values, and thus one cannot judge of their “lack”

Round 2

Reviewer 1 Report

Given the short revision period, I am happy that the authors have made substantial efforts to address my comments and am pleased that the extra data generated adds to the apparent phenotype of reduced GC signaling and increased inflammation in skeletal muscle.

I am therefore happy to recommend the manuscript for publication.